# Sequential Dexterity: Chaining Dexterous Policies for Long-Horizon Manipulation

**Yuanpei Chen**[*], **Chen Wang**[*], **Li Fei-Fei, C. Karen Liu**

Stanford University

**Abstract:** Many real-world manipulation tasks consist of a series of subtasks that are significantly different from one another. Such long-horizon, complex tasks highlight the potential of dexterous hands, which possess adaptability and versatility, capable of seamlessly transitioning between different modes of functionality without the need for re-grasping or external tools. However, the challenges arise due to the high-dimensional action space of dexterous hand and complex compositional dynamics of the long-horizon tasks. We present Sequential Dexterity, a general system based on reinforcement learning (RL) that chains multiple dexterous policies for achieving long-horizon task goals. The core of the system is a *transition feasibility function* that progressively finetunes the sub-policies for enhancing chaining success rate, while also enables autonomous policy-switching for recovery from failures and bypassing redundant stages. Despite being trained only in simulation with a few task objects, our system demonstrates generalization capability to novel object shapes and is able to zero-shot transfer to a real-world robot equipped with a dexterous hand. Code and videos are available at sequential-dexterity.github.io.

**Keywords:** Dexterous Manipulation, Long-Horizon Manipulation, Reinforcement Learning

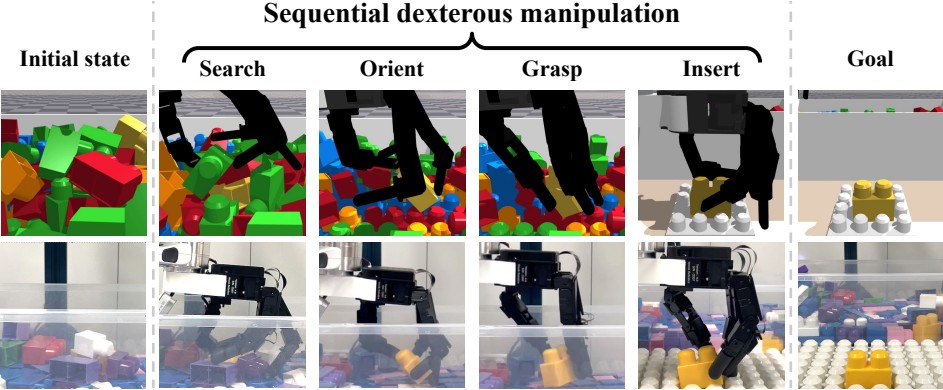

Figure 1: We present **Sequential Dexterity**, a system that learns to chain multiple versatile dexterous manipulation motions for tackling long-horizon tasks (e.g., building a block structure from a pile of blocks), which is able to zero-shot transfer to the real world.

## 1    Introduction

Many real-world manipulation tasks consist of a sequence of smaller but drastically different subtasks. For example, in the task of Lego structure building (Fig.1), the task involves *searching* within a box to locate a specific block piece. Once found, the piece is then *oriented* and *grasped* firmly in hand, setting it up for the final *insertion* at the goal location. Such a task demands a flexible and versatile

---

[*]Equal contribution. Correspondence to Chen Wang <chenwj@stanford.edu>

7th Conference on Robot Learning (CoRL 2023), Atlanta, USA.

manipulator to adapt and switch between different modes of functionality seamlessly, avoiding re-grasping or use of external tools. Furthermore, it requires a long-horizon plan that considers the temporal context and functional relationship between the subtasks in order to successfully execute the entire sequence of tasks. These requirements motivate the use of dexterous hand, which has the potential to reach human-level dexterity by utilizing various hand configurations and their inherent capabilities. However, fully utilizing dexterous hands to achieve long-horizon, versatile tasks remains an outstanding challenge, calling for innovative solutions.

Recent developments in dexterous manipulation have made significant strides in areas such as object grasping [1–3] and in-hand manipulation [4–9]. However, these works primarily investigate single-stage skills, overlooking the potential of sequencing multiple dexterous policies for long-horizon tasks. A naive way to chain multiple dexterous policies together is to simply execute a single-stage skill one after the other. While the simple strategy works in some scenarios [10–12], a subtask in general can easily fail when encountering a starting state it has never seen during training. Regularizing the state space between neighboring skills can mitigate this out-of-distribution issue [13, 14], but long-horizon dexterous manipulation requires a comprehensive optimization of the entire skill chain, due to the complex coordination between non-adjacent tasks. For instance, as depicted in Fig. 1, the robot needs to strategize in advance when orienting the block, aiming for an optimal object pose that facilitates not only the immediate subsequent grasping but also the insertion task in the later stage of the task.

This paper proposes a new method to effectively chain multiple high-dimensional manipulation policies via a combination of regularization and optimization. We introduce a bi-directional process consisting of a forward initialization process and a backward fine-tuning process. The forward initialization process models the end-state distribution of each sub-policy, which defines the initial state distribution for the subsequent policy. The forward initialization process associates the preceding policy with the subsequent one by injecting a bias in the initial state distribution of the subsequent policy during training. Conversely, we also introduce a backward fine-tuning mechanism to associate the subsequent policy with the preceding one. We define a *Transition Feasibility Function* which learns to identify initial states from which the subsequent policy can succeed its task. The transition feasibility function is used to fine-tune the preceding policy, serving as an auxiliary reward signal. With the backward fine-tuning process, the transition feasibility function effectively backpropagates the long-term goals to influence the earlier policies via its learning objectives, thereby enabling global optimization across the entire skill chain. Once the policies are trained and deployed, the transition feasibility functions can be repurposed to serve as stage identifiers that determine the appropriate timing for policy switching and which subsequent policy to switch to. The transition feasibility function substantially improves the robustness of task execution and increases the success rate. Our experimental results demonstrate that the bi-directional optimization process notably enhances the performance of chaining multiple dexterous manipulation policies, which can further zero-shot transfer to a real-world robot arm equipped with a dexterous hand to tackle challenging long-horizon dexterous manipulation tasks.

In summary, the primary contributions of this work encompass:

- The first to explore policy chaining for long-horizon dexterous manipulation.
- A general *bi-directional optimization* framework that effectively chains multiple dexterous skills for long-horizon dexterous manipulation.
- Our framework exhibits state-of-the-art results in multi-stage dexterous manipulation tasks and facilitates zero-shot transfer to a real-world dexterous robot system.

## 2   Related Work

**Dexterous manipulation.**   Dexterous manipulation represents a long-standing area of research in robotics [15–19]. With its high degree of freedom, a dexterous hand can execute a variety of manipulation skills [1–7, 9, 20–24]. Traditional algorithms have typically addressed these challenges by leveraging trajectory optimization founded on analytical dynamics modeling [17–19]. These techniques pose simplification over the active contacts between the hand and objects, limiting their effectiveness in more complex tasks. Conversely, deep reinforcement learning have exhibited the capacity to learn dexterous skills without assumptions for simplification [7, 8, 23]. Despite their

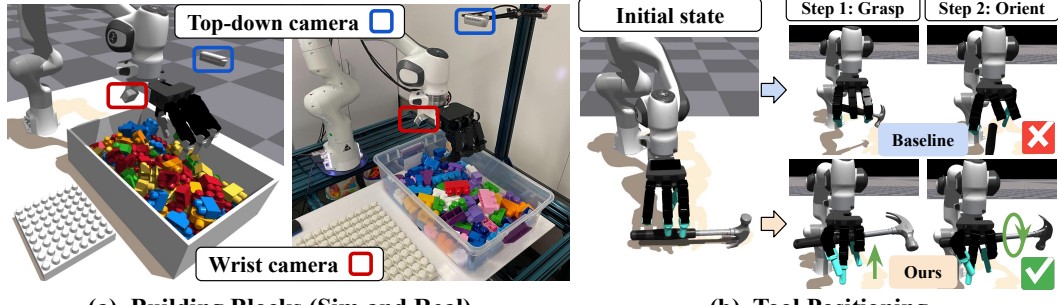

**(a). Building Blocks (Sim and Real)**       **(b). Tool Positioning**

Figure 2: Overview of the environment setups. (a) Workspace of **Building Blocks** task in simulation and real-world. (b) The setup of the **Tool Positioning** task. Initially, the tool is placed on the table in a random pose, and the dexterous hand needs to grasp the tool and re-orient it to a ready-to-use pose. The comparison results illustrate how the way of grasping directly influences subsequent orientation.

notable flexibility in learning dexterous primitives, these methods predominantly focus on singular manipulation tasks such as object re-orientation [5, 6, 25–27] or isolated skills for reset-free learning system [28]. Our work prioritizes the chaining of multiple dexterous primitives, which incorporates the skill feasibility into a comprehensive learning framework for long-horizon dexterous manipulation.

**Long-horizon robot manipulation.** Training a robot to perform long-horizon manipulation tasks from scratch is challenging, primarily due to the cumulative propagation of errors throughout the task execution process. Established methods tackle these tasks by breaking them down into simpler, reusable subtasks [29]. Typically, these algorithms comprise a set of low-level sub-policies, which can be obtained through various means, such as unsupervised exploration [30–34], learning from demonstration [35–39] and pre-defined measures [11, 40–44]. Despite their distinct merits, these works do not address the specific challenge of long-horizon manipulation in the context of dexterous hands. This challenge largely stems from the compounded complexity produced by the extensive state space of a hand coupled with the extended scope of long-horizon tasks. Therefore, even when provided with high-level plans, ensuring a seamless transition between dexterous policies remains a formidable challenge.

**Skill-chaining.** Prior policy-chaining methods focus on updating each sub-task policy to encompass the terminal states of the preceding policy [10, 13]. However, given the high degree of freedom characteristic of a hand, the terminal state space undergoes limitless expansion, thereby complicating effective training. Closely related to our work is Lee et al. [14], wherein a discriminator is learned to regulate the expansion of the terminal state space. Nevertheless, its uni-directional training process restricts optimization to adjacent skills only, disregarding the influence of long-term goals on early non-adjacent policies. In contrast, our bi-directional training mechanism enables the backpropagation of the long-term goal reward to optimize the entire policy chain. Our concept of backward fine-tuning draws significant inspiration from goal-regression planning in classical symbolic planning literatures [45] (also known as pre-image backchaining [46–50]). However, these works assume access to a set of pre-defined motion primitives, which is hard to obtain in dexterous manipulation setups. Our work focuses on the learning and chaining of a sequence of dexterous policies from scratch, targeting the accomplishment of long-horizon task objectives.

## 3 Problem Setups

We study the task of chaining a sequence of dexterous policies to accomplish long-horizon manipulation tasks, examples of which include Lego-like building blocks or picking up and positioning a tool to a desired pose. These two tasks both require the use of multiple dexterous skills to complete, making them highly suitable for studying long-horizon dexterous manipulation with skill chaining.

**Constructing a structure of blocks.** This long-horizon task includes four different subtasks: **searching** for a block with desired dimension and color from a pile of cluttered blocks, **orienting** the block to a favorable position, **grasping** the block, and finally **inserting** the block to its designated position on the structure. This sequence of actions repeats until the structure is completed according to the given assembly instructions. The block set, initially arranged in a random configuration, comprises

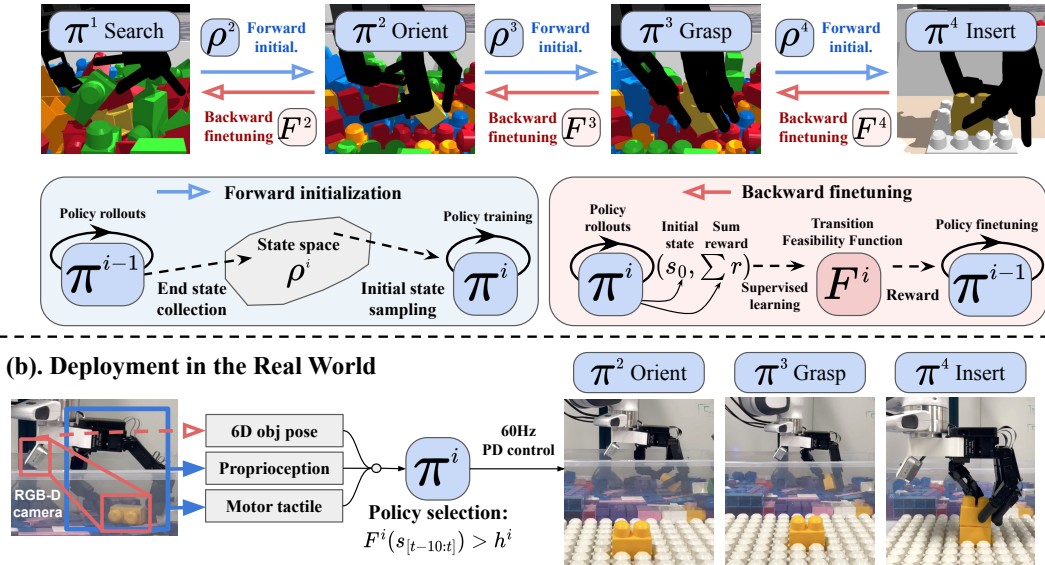

Figure 3: Overview of **Sequential Dexterity**. (a) A bi-directional optimization scheme consists of a forward initialization process and a backward fine-tuning mechanism based on the transition feasibility function. (b) The learned system is able to zero-shot transfer to the real world. The transition feasibility function serves as a policy-switching identifier to select the most appropriate policy to execute.

eight distinct types (different shapes, masses and colors), totaling 72 blocks. We operate under the assumption of having access to an assembly manual that outlines the sequence and desired positioning of each block piece on the board. The environment provides the robot with two RGB-D camera views—one from a top-down camera over the box and the other from the wrist-mounted camera (as is shown in Fig. 2(a)). No other sensors are used in either simulation or the real world. More details of the definition of the sub-task are introduced in Sec. 4.4 and Sec. 5.1.

**Tool positioning.** This task involves two subtasks: **grasping** a tool with a long handle (e.g., hammer, spatula) from a table and **in-hand orienting** it to a ready-to-use pose (e.g., make the flat side of the hammerhead face the nail, as is shown in Fig. 2(b)). The environment provides the robot with the 6D pose of the target tool. For more details on the task setups, please refer to Appendix. F.

## 4 Sequential Dexterity

We propose a bi-directional optimization process to tackle long-horizon dexterous manipulation tasks. Our approach contains three main components: (1) Training dexterous sub-policies (Sec. 4.1), (2) Chaining sub-policies through fine-tuning (Sec. 4.2), (3) Improving system robustness through automatic policy-switching (Sec. 4.3).

### 4.1 Learning dexterous sub-policies

Training a dexterous manipulation policy from scratch for solving long-horizon tasks, like building a block tower (Fig. 1), is significantly challenging given the high degree of freedom associated with a dexterous hand (evidenced by the result of RL-scratch in Tab. 1 and Tab. 2). As such, we first decompose a long-horizon task into a $K$-step sequence of sub-tasks $G = (g^1, g^2, ..., g^K)$ and train each sub-policy $\pi^i$ with Proximal Policy Optimization (PPO) [51] algorithm. We formulate each sub-task as a Markov Decision Process (MDP) $\mathcal{M} = (\boldsymbol{S}, \boldsymbol{A}, \pi, \mathcal{T}, R, \gamma, \rho)$, with state space $\boldsymbol{S}$, action space $\boldsymbol{A}$, policy of the agent $\pi$, transition distribution $\mathcal{T}(\boldsymbol{s}_{t+1}|\boldsymbol{s}_t, \boldsymbol{a}_t)$ ($\boldsymbol{s}_t \in \boldsymbol{S}$, $\boldsymbol{a}_t \in \boldsymbol{A}$), reward function $R$, discount factor $\gamma \in (0,1)$, and initial state distribution $\rho$. The policy $\pi$ outputs a distribution of motor actions $\boldsymbol{a}_t$ based on the current state inputs $\boldsymbol{s}_t$. The goal is to train the policy $\pi$ to maximize the sum of rewards $\mathbb{E}_\pi[\sum_{t=0}^{T-1} \gamma^t r_t]$ ($r_t = R(\boldsymbol{s}_t, \boldsymbol{a}_t, \boldsymbol{s}_{t+1})$) in an episode with $T$ time steps.

However, due to the large state space of a dexterous hand, it is difficult to accurately sample the potential initial states for training individual sub-policies. Take the insertion task as an example (Fig. 3

sub-policy $\pi^4$), randomly sampling the initial hand configuration and object's in-hand pose does not assure a physically stable grasp. However, we provide a critical observation that the successful end state of prior sub-task $\pi^{i-1}$ inherently provides plausible initial states for $\pi^i$ to start with. Similar observation is find in [7]. Inspired by this, we propose a forward initialization training scheme (Fig. 3 (a)). Given a long-horizon task $G = (g^1, g^2, ..., g^K)$, our framework sequentially trains the sub-policies according to the task's chronological order. After training each sub-policy $\pi^i$, we start policy rollouts and collect a set of successful terminal states $\{s_T^i\}$, which is later used as the initial state distribution $\rho^{i+1}$ for training the succeeding policy $\pi^{i+1}$. Such forward training method ensures the validity of the initial states and makes the learning of dexterous policies effective. More details of training sub-policies can be found in Sec. 4.4.

## 4.2 Policy chaining with transition feasibility function

Chaining multiple policies using forward initialization alone may not guarantee success since the previous policy $\pi^{i-1}$ might reach a termination state that its successor $\pi^i$ cannot solve. This issue arises because the preceding policy $\pi^{i-1}$ does not take into account whether the end states are *feasible* for the subsequent policy $\pi^i$ to succeed. To address this challenge, it is crucial to convey the feasibility of the following policy $\pi^i$ *in reverse* to its predecessor $\pi^{i-1}$, enabling the latter to optimize toward states that $\pi^i$ can handle. Based on this hypothesis, we propose a backward policy fine-tuning mechanism with a transition feasibility function (Fig. 3 (b)).

**Learning transition feasibility function.** The feasibility of a given state for a policy can be described as the policy's ability to succeed in the end when starting from that state. We formalize this concept by creating a function that maps the transition state $s_0^i \in \rho^i$ (which is equivalent to $s_T^{i-1}$) to the expected sum of reward within the sub-task execution, $\mathbb{E}_{\pi^i}[\sum_{t=0}^{T-1} r_t]$. We name this function, $F : \mathcal{S} \mapsto \mathbb{R}$, the *Transition Feasibility Function*. However, a single state $s_T^{i-1}$ is not sufficient to differentiate the performance of $\pi^i$. In particular, the velocity of object from the previous sub-task may be critical to the performance of $\pi^i$, which cannot be captured by $s_T^{i-1}$ alone. As a result, the transition feasibility function takes a sequence of observation states $s_{[T-10:T]}^{i-1}$ (10 steps in our experiments) as input and employs a multi-head attention network [52] to extract suitable temporal information for learning the $F$. The final learning objective of the transition feasibility function $F^i$ for sub-policy $\pi^i$ is:

$$\mathcal{L}^i = \|F^i(s_{[T-10:T]}) - \mathbb{E}_{\pi^i}[\sum_{t=0}^{T-1} r_t]\|_2 \tag{1}$$

**Backward policy fine-tuning.** Once $F^i$ is trained, we can fine-tune the prior policy $\pi^{i-1}$, by incorporating $F^i$ as an auxiliary reward component. The fine-tuning starts with updating the second-to-the-last sub-task policy $\pi^{K-1}$ and sequentially moves backward, refining each preceding policy until the first one $\pi^1$ is updated. In each fine-tuning step, we utilize $F^i$ as an additional reward, combined with the original sub-task reward $R^{i-1}$, to fine-tune policy $\pi^{i-1}$. The final policy fine-tuning reward function is:

$$R^{i-1'}(s_t, a_t, s_{t+1}; F^i) = \lambda_1 R^{i-1}(s_t, a_t, s_{t+1}) + \lambda_2 F^i(s_{[T-10:T]}), \tag{2}$$

where $\lambda_1$ and $\lambda_2$ are the weighting factors. Once $\pi^{i-1}$ has been refined, we execute policy rollouts to gather data, which maps from the initial state $s_{[T-10:T]}^{i-2}$ to the accumulated reward $\mathbb{E}_{\pi^{i-1}}[\sum_{t=0}^{T-1} r_t]$ received by the policy $\pi^{i-1}$ at the terminal state $s_T^{i-1}$. This data helps to construct a new transition feasibility function $F^{i-1}$, which is further used to fine-tune the preceding policy $\pi^{i-2}$. The implementation pseudocode of the bi-directional forward and backward training process is illustrated in Appendix. A.

## 4.3 Policy switching with transition feasibility function

A key challenge in chaining multiple dexterous policies is to determine *when* to switch to the next policy and *what* should be the next policy to execute. Prior works approach this issue by establishing a predetermined execution horizon for a policy, transitioning abruptly to the subsequent policy once the maximum step count is attained [10, 14, 53–55]. The pre-scheduled policy transitions worked in some scenarios, but they are not suitable for dexterous manipulation that involves non-prehensile

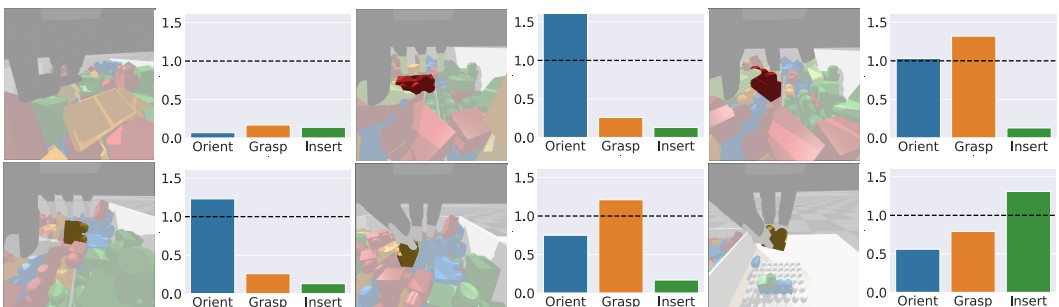

Figure 4: Examples of policy-switching with transition feasibility function. Each example contains an image from the wrist-mount camera (left) and its corresponding feasibility score $c_i$ outputted by the transition feasibility function (right). We highlight the target block in the image for better visualization. The policy-switching process visits each sub-policy in reverse order. The first sub-policy with a feasibility score $c_i > 1.0$ is selected for execution.

maneuvers and in-hand manipulation. For instance, if a robot is reorienting an object in-hand, a premature policy switch before the object is stabilized could result in task failure. The key to tackling this issue is to automatically figure out the appropriate switch time such that the transition state will lead to success of next policy. The transition feasibility function provides exactly the information we need for identifying the switch timing. As such, during execution, we repurpose our trained transition feasibility functions as a policy-switching identifier. At each time step, the transition feasibility function of the next sub-policy will output a feasibility score $c_t^{i+1} = F^{i+1}(\boldsymbol{s}_{[t-10:t]})/h^{i+1}$, where $h^{i+1}$ is a threshold hyperparameter defined based on the reward of successful task executions. The ideal time of policy switching can then be defined as the moment when $c_t^{i+1} > 1$.

Simply executing sub-policies sequentially may not guarantee successful task execution since the robot sometimes needs to recover using a previous policy and sometimes needs to bypass a future policy if the sub-task has already been achieved. Thus, effective policy switching requires the robot to not only consider the current policy and its successor, but also the entire skill chain. To achieve this, we group the learned transition feasibility function $(F^2, F^3..., F^K)$ as a stage estimator. At each policy-switching step, we calculate the feasibility score from the final transition feasibility function of the entire task $c_t^K = F^K(\boldsymbol{s}_{[t-10:t]})/h^K$, sequentially moving backward. The sub-policy, for which the first feasibility score $c_t^i > 1$, is considered the next policy for execution. If none of the feasibility scores satisfy, the robot will restart from the beginning of the entire task. Leveraging the learned transition feasibility function in this manner enhances the robot's robustness against unexpected failures during policy execution, while also allowing it to bypass redundant stages, thus promoting efficient task execution.

### 4.4 Implementation details

**RL reward.** Training sub-policies require pre-defined sub-task rewards $\{R^i\}_{i=1}^K$. Establishing such rewards can be complex as the appropriate sub-goals that would most contribute to the overall task accomplishment may not be readily apparent. However, we pose a critical finding that the backward fine-tuning mechanism can transmit the goal of the entire task to each sub-task. For instance, the transition feasibility function of the inserting policy informs the grasping policy about the in-hand object pose that would be most beneficial for the insertion. Furthermore, such backward transmission can influence all preceding sub-policies, enabling the entire policy chain to optimize for the overall task goal. This mechanism alleviates the burden of reward shaping and allows us to use standard sub-task rewards that are agnostic to the final task goal for training sub-policies. For instance, the sub-task reward of grasping is defined as whether the target object has been lifted. The specifics of how the object is held in hand are automatically managed during the backward fine-tuning process. The detailed descriptions of each sub-task reward are documented in the Appendix. D.

**State-action space.** The state space for the sub-policies is built around the perspective of the hand. It integrates proprioception and motor tactile [56, 57] information from the 16-degree-of-freedom Allegro Hand as well as the target object's 6D pose in the reference frame of the wrist-mounted camera. During simulation-based sub-policy training, we augment this state space with additional information,

| | Trained | | | | | Unseen | | | |
| | Block 1 | Block 2 | Block 3 | Block 4 | Block 5 | Block 6 | Block 7 | Block 8 | ALL |
|---|---|---|---|---|---|---|---|---|---|
| RL-scratch | 0.00±0.00 | 0.00±0.00 | 0.00±0.00 | 0.00±0.00 | 0.00±0.00 | 0.00±0.00 | 0.00±0.00 | 0.00±0.00 | 0.00±0.00 |
| Curriculum RL | 0.00±0.00 | 0.00±0.00 | 0.00±0.00 | 0.00±0.00 | 0.00±0.00 | 0.00±0.00 | 0.00±0.00 | 0.00±0.00 | 0.00±0.00 |
| V-Chain [34] | 0.15±0.02 | 0.09±0.04 | 0.11±0.04 | 0.10±0.04 | 0.08±0.02 | 0.08±0.03 | 0.03±0.02 | 0.04±0.02 | 0.08±0.02 |
| Policy-Seq [10] | 0.20±0.04 | 0.14±0.02 | 0.15±0.03 | 0.23±0.03 | 0.15±0.03 | 0.17±0.00 | 0.16±0.01 | 0.12±0.02 | 0.16±0.02 |
| T-STAR [14] | 0.19±0.04 | 0.18±0.02 | 0.11±0.01 | 0.27±0.02 | 0.17±0.04 | 0.25±0.02 | **0.26±0.03** | 0.10±0.03 | 0.19±0.03 |
| Ours w/o temporal | 0.47±0.06 | 0.44±0.07 | 0.43±0.04 | 0.49±0.04 | 0.40±0.04 | 0.51±0.04 | 0.18±0.01 | **0.16±0.03** | 0.38±0.04 |
| Ours | **0.61±0.03** | **0.55±0.01** | **0.52±0.03** | **0.63±0.03** | **0.51±0.06** | **0.53±0.06** | 0.22±0.02 | **0.16±0.01** | **0.46±0.03** |

Table 1: Results for the Building Blocks task

such as the velocity of each hand joint and the target object. In real-world deployments, these states are abstracted via policy distillation [6, 7, 58]. The action space of our system includes 16-dimensional hand joints and 3D wrist translation of the robot arm. To concentrate the sub-policy learning on critical aspects of the manipulation task, when the target's (either object or goal location) relative pose to the wrist camera is detected more than 5 centimeters from the hand, we employ a motion-planning-based operational space controller (OSC [59]) to move the end-effector to a position 5 centimeters above the target. More details of the state-action space are introduced in Appendix. C.

# 5 Experiments

## 5.1 Experiment setups

**Environment setups.** The environment is initialized with a Franka Emika robot arm equipped with an Allegro Hand as the end-effector. In the Building Block task, we placed a box of blocks (eight categories, in total 72 pieces) and a building board on the table. For the tool positioning task, a long-handled tool is placed on the table. The control frequency is 60 Hz for both the robot arm and the hand. The real-world hardware mirrors the simulation setups. We coordinate the use of top-down and wrist-mount cameras to access the 6D pose and segmentation mask of the object in the real-world: we use the top-down camera once at the beginning, use the wrist-mount camera once at the beginning of each maneuver in **orienting**, and use the wrist-mount camera continuously in **searching**, **grasping** and **inserting**; More details of the environment setups are in Appendix. F.

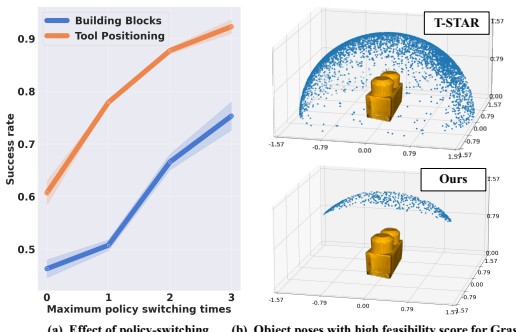

(a). Effect of policy-switching  (b). Object poses with high feasibility score for Grasp

Figure 5: (a) Performance improvement of Ours given 0/1/2/3 maximum policy-switching times. (b) Visualization of object poses with high feasibility score for the Grasp sub-policy in Building Blocks task. The x, y, and z axes are the roll, yaw, and pitch of the object, respectively. In Ours, each point in the diagram represents a pose that is regarded as feasible by the transition feasibility function ($c^i > 1.0$). For T-STAR, we use the poses that are judged as successful by its discriminator.

**Baseline methods.** We compare our approach with the following baselines: 1) RL-scratch is vanilla PPO algorithm [51] learns the task from scratch. 2) Curriculum RL follows a procedure training strategy to expand from the first skill to the entire task. 3) V-Chain [34] combines skill-chaining with the value function from the PPO policy. 4) Policy-Seq [10] focuses on the forward initiation process in skill-chaining. 5) T-STAR [14] incorporates a discriminator to regularize the terminal states.

## 5.2 Results

**Bi-directional optimization framework is key for chaining multiple dexterous policies.** In Tab. 1 and Tab. 2, our approach learned with bi-directional optimization (Ours and Ours w/o temporal) outperforms prior uni-directional skill-chaining methods (V-Chain, Policy-Seq and T-STAR) significantly, with more than 20% improvement in task success rate in two long-horizon tasks. We further analyze what really matters for successful policy chaining. We visualize the transition feasibility score of the *grasping* sub-policy (T-STAR's result is calculated from its discriminator) in Fig. 5(b). We found our approach with backward fine-tuning scheme correctly transits the goal of the succeeding *inserting* skill to prior *grasping* and encourages the policy to grasp the block when its studs face up, which facilitates

| | Trained Hammer | Unseen Spatula | Unseen Spoon | ALL |
|---|---|---|---|---|
| RL-scratch | $0.17_{\pm 0.05}$ | $0.06_{\pm 0.03}$ | $0.10_{\pm 0.01}$ | $0.11_{\pm 0.03}$ |
| Curriculum RL | $0.29_{\pm 0.02}$ | $0.17_{\pm 0.01}$ | $0.16_{\pm 0.08}$ | $0.21_{\pm 0.04}$ |
| Policy-Seq [10] | $0.43_{\pm 0.01}$ | $0.29_{\pm 0.06}$ | $0.24_{\pm 0.04}$ | $0.32_{\pm 0.02}$ |
| T-STAR [14] | $0.47_{\pm 0.01}$ | $0.40_{\pm 0.03}$ | $0.26_{\pm 0.04}$ | $0.37_{\pm 0.03}$ |
| Ours w/o temp. | $0.77_{\pm 0.03}$ | $0.54_{\pm 0.07}$ | $0.40_{\pm 0.04}$ | $0.57_{\pm 0.05}$ |
| Ours | $\mathbf{0.81}_{\pm 0.01}$ | $\mathbf{0.57}_{\pm 0.04}$ | $\mathbf{0.43}_{\pm 0.08}$ | $\mathbf{0.60}_{\pm 0.04}$ |

Table 2: Results for the tool positioning task

| | Single Block 1 | Single Block 4 | Double Block 1 |
|---|---|---|---|
| RL-scratch | 0/10 | 0/10 | 0/10 |
| Policy-Seq [34] | 0/10 | 2/10 | 0/10 |
| T-STAR [14] | 3/15 | 5/18 | 0/13 |
| Ours | **12/20** | **10/20** | **5/15** |

Table 3: Real world results in the Building Blocks task. Single/Double refers to building one single block or stacking two blocks.

the insertion. T-STAR with the uni-directional learning process, however, suggests many states where the block's studs are facing horizontally, thereby bringing challenges for subsequent insertion.

**Transition feasibility function significantly improves performance of long-horizon dexterous manipulation.** In Tab. 1 and Tab. 2, the models learned with the transition feasibility function (Ours and Ours w/o temporal) outperforms the one using the PPO-trained value function (V-chain) for more than 30% in task success rate. This result implies that the value function of PPO policy fails to model the feasibility of subsequent policy, which further affects policy chaining results.

**Temporal inputs facilitate handling high-dimensional state spaces, particularly for dexterous manipulation.** In Tab. 1, by training the transition feasibility function to extract temporal information from a sequence of history states, Ours exceeds Ours w/o temporal for 8% in task success rate. This result highlights the importance of extracting velocity and temporal information for chaining dexterous policies that contain dynamic finger motions.

**Ability to switch sub-policy autonomously is essential for succeeding long-horizon tasks.** Fig. 5(a) illustrates the performance improvement of enabling automatic policy-switching. Only with a maximum allowance of three switching times, Ours can improve more than 30% in task success rate. This result concludes that it is crucial to have the capability of switching forward and backward by leveraging the transition feasibility function of each sub-policy. Such policy-switching ability further contributes to our real-world results in Tab. 3, which allows the policy to handle a challenging 8-step long-horizon task (Double Block 1) with more than 30% task success rate (10 maximum switching times for all methods in the real-world experiments). Please refer to the website for more results.

**Real-world results.** In Tab. 3 real-world experiments, our approach has more than 30% success rate improvements compared to prior methods. This result is consistent with the results in simulation. Ours has a 33% success rate in building a double-block structure, while the other baselines have a 0% success rate. This result highlights the ability of our model when tackling long-horizon dexterous manipulation tasks. For more details of the real-world setups, please refer to Appendix. B.

## 6 Limitations

There are several limitations of our work. First, we encounter difficulties in simulating a contact-rich insertion process which necessitates an additional manually designed pressing motion to completely insert the blocks during real-world deployment. Second, the motor tactile does not yield a significant improvement in performance, as observed in Appendix Tab. 8. Our future research could explore the potential of sensor-based tactile signals for contact-rich tasks, as proposed in [60, 61, 20, 24, 27].

## 7 Conclusion

We present Sequential Dexterity, a system developed for tackling long-horizon dexterous manipulation tasks. Our system leverages a bi-directional optimization process to chain multiple dexterous policies learned with deep reinforcement learning. At the core of our system is the Transition Feasibility Function, a pivotal component facilitating a gradual fine-tuning of sub-policies and enabling dynamic policy switching, thereby significantly increasing the success rate of policy chaining. Our system has the capability to zero-shot transfer to a real-world dexterous robot, exhibiting generalization across novel object shapes. Our bi-directional optimization framework also has the potential to be a general skill chaining method beyond dexterous manipulation. Potential applications including chaining skills for bimanual robots.

## Acknowledgments

This research was supported by National Science Foundation NSF-FRR-2153854, NSF-NRI-2024247, NSF-CCRI-2120095 and Stanford Institute for Human-Centered Artificial Intelligence, SUHAI. In the real-world experiment, the controller of the Franka Emika Panda arm is developed based on Deoxys [62] and the Allegro hand is controlled through zmq [2]. We would also like to thank Ruocheng Wang, Wenlong Huang, Yunzhi Zhang and Albert Wu for providing feedback on the paper.

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

## A  Training pseudocode

---

**Algorithm 1** SEQUENTIAL DEXTERITY: A bi-directional optimization framework for skill chaining

---

**Require:** sub-task MDPs $\mathcal{M}_1,...,\mathcal{M}_K$

1: Initialize sub-policies $\pi_\theta^1,...,\pi_\theta^K$, transition feasibility function $F_\omega^1,...,F_\omega^K$, ternimal state buffers $\mathcal{B}_\mathcal{T}^1,...,\mathcal{B}_\mathcal{T}^K$, the sum of reward buffers $\mathcal{B}_\mathcal{R}^1,...,\mathcal{B}_\mathcal{R}^K$, and the weighting factors of the backward fine-tuning $\lambda_1$ and $\lambda_2$.

2: **for** iteration $m=0,1,...,M$ **do**

3:     **for** each sub-task $i=1,...,K$ **do**

4:         **while** until convergence of $\pi_\theta^i$ **do**

5:             Rollout trajectories $\tau=(s_0^i,a_0^i,r_0^i,...,s_T^i)$ with $\pi_\theta^i$

6:             Update $\pi_\theta^i$ by maximizing $\mathbb{E}_{\pi^i}[\sum_{t=0}^{T-1}\gamma^t r_t^i]$

7:         **end while**

8:     **end for**                                                   ▷ Forward initialization

9:     **for** each sub-task $i=K,...,2$ **do**

10:         **while** until convergence of $\pi_\theta^{i-1}$ **do**

11:             Rollout trajectories $\tau^{i-1}=(s_0^{i-1},a_0^{i-1},r_0^{i-1},...,s_T^{i-1})$ with $\pi_\theta^{i-1}$

12:             Sample $s_0^i$ from environment or $\mathcal{B}_\beta^{i-1}$

13:             Rollout trajectories $\tau^i=(s_0^i,a_0^i,r_0^i,...,s_T^i)$ with $\pi_\theta^i$

14:             **if** sub-task $i$ is complete **then**

15:                 $\mathcal{B}_\mathcal{T}^i\leftarrow\mathcal{B}_\mathcal{T}^i\cup s_{[T-10:T]},\mathcal{B}_\mathcal{R}^i\leftarrow\mathcal{B}_\mathcal{R}^i\cup[\sum_{t=0}^{T-1}r_t^i]$

16:             **end if**

17:             Update $F^i$ with $s_{[T-10:T]}\sim\mathcal{B}_\mathcal{T}^{i-1}$ and $[\sum_{t=0}^{T-1}r_t]\sim\mathcal{B}_\mathcal{R}^i$

18:             Update $\pi_\theta^{i-1}$ by maximizing $\mathbb{E}_{\pi^{i-1}}[\sum_{t=0}^{T-1}\gamma^t(\lambda_1 R^{i-1}(\boldsymbol{s}_t^{i-1},\boldsymbol{a}_t^{i-1},\boldsymbol{s}_{t+1}^{i-1})+\lambda_2 F_\omega^i(\boldsymbol{s}_{[T-10:T]}^i))]$

19:         **end while**

20:     **end for**                                                   ▷ Backward finetuning

21: **end for**

---

## B  Real-world system setups

During real-world deployment, some observations used in the simulation are hard to accurately estimate (e.g., joint velocity, object velocity, etc.). We use the teacher-student policy distillation framework [6, 7, 58] to abstract away these observation inputs from the policy model. In each policy rollout, our system first uses the top-down camera view to perform a color-based segmentation to localize the target block piece given by the manual. Then, the robot calls motion planning API to move to the target location with OSC controller [59]. After that, our system uses the wrist camera view to track the segmentation and 6D pose of the object with a combination of color-based initial segmentation, Xmem segmentation tracker [63], and Densefusion pose estimator [64]. If the target object is deeply buried (as the case in the top left corner of Fig. 4), the transition feasibility function will inform the robot to execute the searching policy until the target appears. During the last insertion stage, the estimated 6D object pose will guide the robot policy to adjust its finger and wrist motion to align with the goal location as it learned in the simulation. Since simulating contact-rich insertion is still a research challenge in graphics, after the robot has placed the block to the target location, we perform a scripted pressing motion (spread out the entire hand and press down) on the target location to ensure a firm insert. The output of the policy which controls the hand is low-pass filtered with an exponential moving average (EMA) smoothing factor [6], which can also effectively reduce jittering motions. Our results in the real-world were obtained with an EMA of 0.2, which provides a balance between agility and stability of the motions. More details about real-world system setups and results can be found in the Supplementary video.

## C  State Space in Simulation

### C.1  Building Blocks

**Searching**  Table.4 gives the specific information of the state space of the searching task.

Table 4: Observation space of Search task.

| Index | Description |
| --- | --- |
| 0 - 23 | dof position |
| 23 - 46 | dof velocity |
| 46 - 98 | fingertip pose, linear velocity, angle velocity (4 x 13) |
| 98 - 111 | hand base pose, linear velocity, angle velocity |
| 111 - 124 | object base pose, linear velocity, angle velocity |
| 124 - 143 | the actions of the last timestep |
| 143 - 159 | motor tactile |
| 159 - 160 | the number of pixels occupied by the target object mask |

**Orienting**  Table.5 gives the specific information of the state space of the orienting task.

Table 5: Observation space of Orient and Grasp task.

| Index | Description |
| --- | --- |
| 0 - 23 | dof position |
| 23 - 46 | dof velocity |
| 46 - 98 | fingertip pose, linear velocity, angle velocity (4 x 13) |
| 98 - 111 | hand base pose, linear velocity, angle velocity |
| 111 - 124 | object base pose, linear velocity, angle velocity |
| 124 - 143 | the actions of the last timestep |
| 143 - 159 | motor tactile |

**Grasping**  Table.5 gives the specific information of the state space of the grasping task.

**Inserting**  Table.6 gives the specific information of the state space of the inserting task.

Table 6: Observation space of Insert task.

| Index | Description |
| --- | --- |
| 0 - 23 | dof position |
| 23 - 46 | dof velocity |
| 46 - 98 | fingertip pose, linear velocity, angle velocity (4 x 13) |
| 98 - 111 | hand base pose, linear velocity, angle velocity |
| 111 - 124 | object base pose, linear velocity, angle velocity |
| 124 - 143 | the actions of the last timestep |
| 143 - 159 | motor tactile |
| 159 - 166 | goal pose |
| 166 - 169 | goal position - object position |
| 169 - 173 | goal rotation - object rotation |

### C.2  Tool positioning

**Grasping**  Table.5 gives the specific information of the state space of the grasping task.

**In-hand Orientation**  Table.6 gives the specific information of the state space of the in-hand orientation task.

Table 7: Domain randomization of all the sub-tasks.

| Parameter | Type | Distribution | Initial Range |
|---|---|---|---|
| **Robot** | | | |
| Mass | Scaling | uniform | [0.5, 1.5] |
| Friction | Scaling | uniform | [0.7, 1.3] |
| Joint Lower Limit | Scaling | loguniform | [0.0, 0.01] |
| Joint Upper Limit | Scaling | loguniform | [0.0, 0.01] |
| Joint Stiffness | Scaling | loguniform | [0.0, 0.01] |
| Joint Damping | Scaling | loguniform | [0.0, 0.01] |
| **Object** | | | |
| Mass | Scaling | uniform | [0.5, 1.5] |
| Friction | Scaling | uniform | [0.5, 1.5] |
| Scale | Scaling | uniform | [0.95, 1.05] |
| Position Noise | Additive | gaussian | [0.0, 0.02] |
| Rotation Noise | Additive | gaussian | [0.0, 0.2] |
| **Observation** | | | |
| Obs Correlated. Noise | Additive | gaussian | [0.0, 0.001] |
| Obs Uncorrelated. Noise | Additive | gaussian | [0.0, 0.002] |
| **Action** | | | |
| Action Correlated Noise | Additive | gaussian | [0.0, 0.015] |
| Action Uncorrelated Noise | Additive | gaussian | [0.0, 0.05] |
| **Environment** | | | |
| Gravity | Additive | normal | [0, 0.4] |

# D  Reward functions

## D.1  Building Blocks

**Searching**  Denote the $\tau$ is the commanded torques at each timestep, the count of individual pixels within the target object's segmentation mask in the top-down camera frame as $\boldsymbol{P}$, The sum of the distance between each fingertip and the object as $\sum_{i=0}^{4} \boldsymbol{f}_i$, the action penalty as $\|\boldsymbol{a}\|_2^2$, and the torque penalty as $\|\tau\|_2^2$. Finally, the rewards are given by the following specific formula:

$$r = \lambda_1 * \boldsymbol{P} + \lambda_2 * min(e_0 - \sum_{i=0}^{4} \boldsymbol{f}_i, 0) + \lambda_3 * \|\boldsymbol{a}\|_2^2 + \lambda_4 * \|\tau\|_2^2 \tag{3}$$

where $\lambda_1 = 5.0$, $\lambda_2 = 1.0$, $\lambda_3 = -0.001$, $\lambda_4 = -0.003$, and $e_0 = 0.2$.

**Orienting**  Denote the $\tau$ is the commanded torques at each timestep, the angular distance between the current object pose and the initial pose as $\theta$, the sum of the distance between each fingertip and the object as $\sum_{i=0}^{4} \boldsymbol{f}_i$, the action penalty as $\|\boldsymbol{a}\|_2^2$, and the torque penalty as $\|\tau\|_2^2$. Finally, the rewards are given by the following specific formula:

$$r = \lambda_1 * \theta + \lambda_2 * min(e_0 - \sum_{i=0}^{4} \boldsymbol{f}_i, 0) + \lambda_3 * \|\boldsymbol{a}\|_2^2 + \lambda_4 * \|\tau\|_2^2 \tag{4}$$

where $\lambda_1 = 1.0$, $\lambda_2 = 1.0$, $\lambda_3 = -0.001$, $\lambda_4 = -0.003$, and $e_0 = 0.6$.

**Grasping**  Denote the $\tau$ is the commanded torques at each timestep, the sum of the distance between each fingertip and the object as $\sum_{i=0}^{4} \boldsymbol{f}_i$, the action penalty as $\|\boldsymbol{a}\|_2^2$, and the torque penalty as $\|\tau\|_2^2$. Finally, the rewards are given by the following specific formula:

$$r = \lambda_1 * exp[\alpha_0 * min(e_0 - \sum_{i=0}^{4} \boldsymbol{f}_i, 0)] + \lambda_2 * \|\boldsymbol{a}\|_2^2 + \lambda_3 * \|\tau\|_2^2 \tag{5}$$

where $\lambda_1 = 1.0$, $\lambda_2 = -0.001$, $\lambda_3 = -0.003$, $\alpha_0 = -5.0$, and $e_0 = 0.1$. It is worth noting that in the latter half of our grasping training, we force the hand to lift, so if the grip is unstable, the object will drop and the reward will decrease.

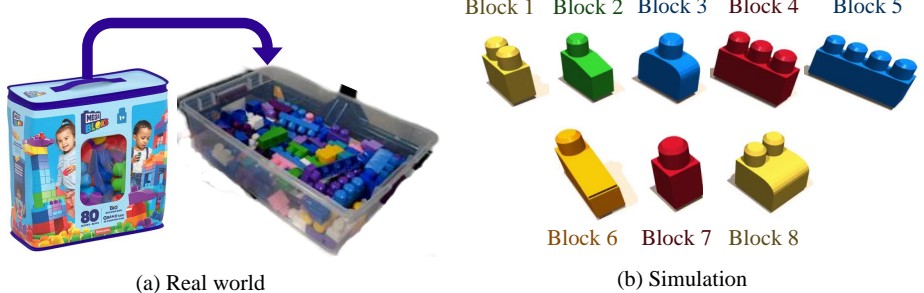

(a) Real world        (b) Simulation

Figure 6: The block model we use in simulation and real-world. (b) is the eight blocks used in our building blocks task. The upper Block 1-5 is the training block, and the lower Block 6-8 is the unseen block for testing.

**Inserting**    Denote the $\tau$ is the commanded torques at each timestep, the object and goal position as $x_o$ and $x_g$, the angular position difference between the object and the goal as $d_a$, the sum of the distance between each fingertip and the object as $\sum_{i=0}^{4} f_i$, the action penalty as $\|a\|_2^2$, and the torque penalty as $\|\tau\|_2^2$. Finally, the rewards are given by the following specific formula:

$$r = \lambda_1 * exp[-(\alpha_0 * \|x_o - x_g\|_2 + \alpha_1 * 2 * \arcsin(clamp(\|d_a\|_2, 0, 1)))] + \\ \lambda_2 * min(e_0 - \sum_{i=0}^{4} f_i, 0) + \lambda_3 * \|a\|_2^2 + \lambda_4 * \|\tau\|_2^2 \tag{6}$$

where $\lambda_1 = 1.0$, $\lambda_2 = 0.0$, $\lambda_3 = -0.001$, $\lambda_4 = -0.003$, $\alpha_0 = 20.0$, $\alpha_1 = 1.0$, and $e_0 = 0.06$.

## D.2    Tool positioning

**Grasping**    Denote the $\tau$ is the commanded torques at each timestep, the sum of the distance between each fingertip and the object as $\sum_{i=0}^{4} f_i$, the action penalty as $\|a\|_2^2$, and the torque penalty as $\|\tau\|_2^2$. Finally, the rewards are given by the following specific formula:

$$r = \lambda_1 * exp[\alpha_0 * min(e_0 - \sum_{i=0}^{4} f_i, 0)] + \lambda_2 * \|a\|_2^2 + \lambda_3 * \|\tau\|_2^2 \tag{7}$$

where $\lambda_1 = 1.0$, $\lambda_2 = -0.001$, $\lambda_3 = -0.003$, $\alpha_0 = -5.0$, and $e_0 = 0.1$. It is worth noting that in the latter half of our grasping training, we force the hand to lift, so if the grip is unstable, the object will drop and the reward will decrease.

**In-hand Orientation**    Denote the $\tau$ is the commanded torques at each timestep, the object and goal position as $x_o$ and $x_g$, the angular position difference between the object and the goal as $d_a$, the sum of the distance between each fingertip and the object as $\sum_{i=0}^{4} f_i$, the action penalty as $\|a\|_2^2$, and the torque penalty as $\|\tau\|_2^2$. Finally, the rewards are given by the following specific formula:

$$r = \lambda_1 * exp[-(\alpha_0 * \|x_o - x_g\|_2 + \alpha_1 * 2 * \arcsin(clamp(\|d_a\|_2, 0, 1)))] + \\ \lambda_2 * min(e_0 - \sum_{i=0}^{4} f_i, 0) + \lambda_3 * \|a\|_2^2 + \lambda_4 * \|\tau\|_2^2 \tag{8}$$

where $\lambda_1 = 1.0$, $\lambda_2 = 0.0$, $\lambda_3 = -0.001$, $\lambda_4 = -0.003$, $\alpha_0 = 20.0$, $\alpha_1 = 1.0$, and $e_0 = 0.06$.

## D.3    Reward Construction

We use an exponential map in the grasping reward function, which is an effective reward shaping technique used in the case to minimize the distance between fingers and object (e.g., grasping task), introduced by [65, 66]. For the same term in the other two reward function, since the other two reward

functions mainly consider other objectives, we empirically find there is no need to use exponential map in these cases. To improve the calculation efficiency, we use quaternion to represent the object orientation. The angular position difference is then computed through the dot product between the normalized goal quaternion and the current object's quaternion.

# E    Domain Randomization

Isaac Gym provides lots of domain randomization functions for RL training. We add the randomization for all the sub-tasks as shown in Table. 7 for each environment. we generate new randomization every 1000 simulation steps.

# F    Task Setups

## F.1    Sub-task definition.

Here we introduce the functionalities of each sub-policy in the Building Block task: **Search** aims to dig and retrieve the target block when it is buried by other blocks in the box. The initial task goal is to make the target block's visible surface in the wrist-view camera larger than a threshold. The transition feasibility function finetunes the policy to a reach a state that facilitates the succeeding orientation. **Orient** aims to rotate the target block. The initial task goal is to freely rotating the target block in-hand without specific goal pose. In the backward step, the transition feasibility function finetunes the policy to rotate the block to a pose that facilitates the succeeding grasping and insertion. **Grasp** aims to lift up the target block and hold in-hand. The initial task goal is to lift up the target block for more than 30 centimeters. The transition feasibility function further finetunes the policy to grasp the block in a way that allows the succeeding insertion to in-hand adjust the block for 90 degrees depending on the given task goal. **Insert** aims to rotate the block in-hand for 90 degrees if the goal pose of the block is vertical and adjust the robot's wrist position with 3D delta motions to align the block with the desired insertion location. In the real-world experiments, since the finger motor of the dexterous hand is not strong enough to fully insert the block, we add an additional scripted pressing motion to complete the last step of the insertion.

## F.2    Building Blocks

**Block model.**    For the building blocks task, we use the same model as Mega Bloks[1] as our blocks. It is a range of large, stackable construction blocks designed specifically for the small hands of the children. We take eight different types of blocks (denoted as Block 1, Block 2,..., Block 8) as the models of our block, and carefully measured the dimensions to ensure that they were the same as in the real world. The block datasets is shown in Figure. 6. For all building block sub-tasks, we use Block 1-5 as the training object and Block 6-8 as the unseen object for testing.

**Physics in insertion between two blocks.**    It is difficult to simulate the realistic insertion in the simulator, and it is easy to explode or model penetration when the two models are in frequent contact. Therefore, we want the plug and slot between the two blocks can be inserted without frequent friction. We reduced the diameter of all block plugs and convex decomposed them via VHACD method when loaded into Isaac Gym. Finally, we made one block possible to insert another block through free fall to verify the final effect.

**Initialization.**    In simulation, we randomly sample the initial block placement above the box, allowing them to fall and form the initial scene. In the real-world experiments, we manually shuffle the blocks' placement in the box, with the shuffling based on the criteria that none of the task-related blocks lies within the margin of 10 centimeters from the edges of the box. If the criteria are not satisfied, we re-shuffle the blocks.

---

[1]https://www.megabrands.com/en-us/mega-bloks.

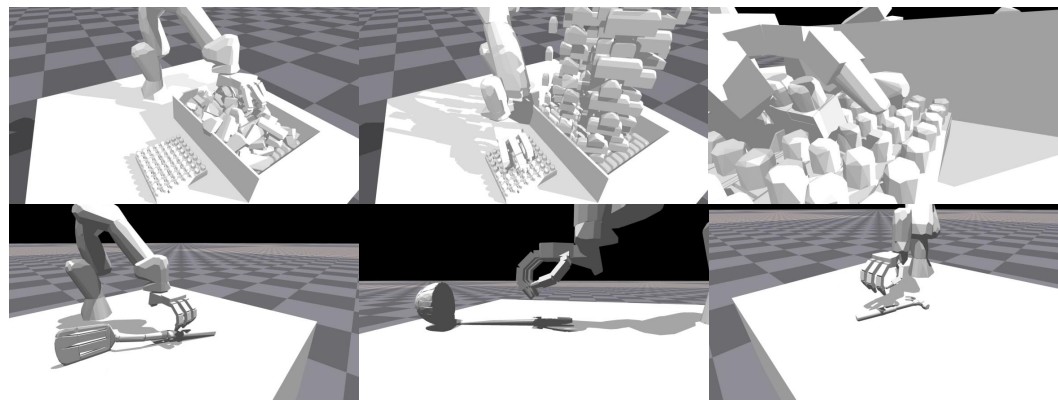

Figure 7: The collision meshes in the simulation.

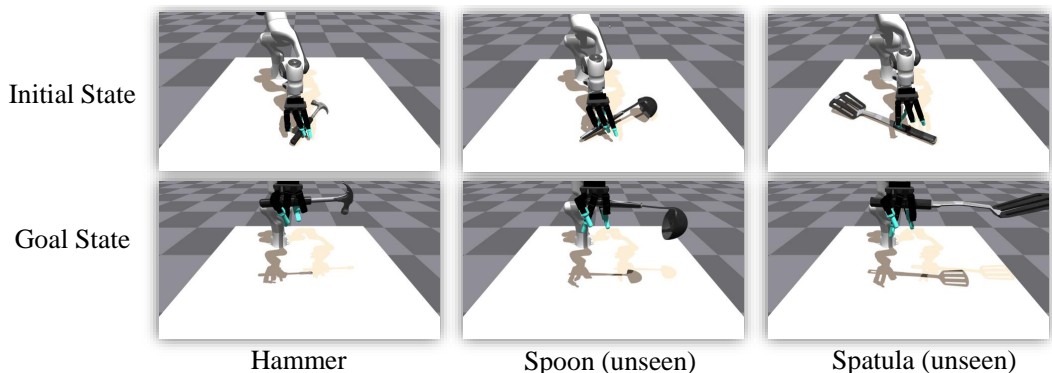

Figure 8: Visualization of the three tools we use in Tool Positioning task. The Hammer is use for training and the Spoon and Spatula is only use for testing. We also show the goal pose of the tools.

**Success criteria.** In the Building Block task, the task success is defined as whether the target block has been inserted on the desired pose on the LEGO board. We assume the access to a building manual that specifies the shape and color of the target block and its desired goal pose on the board. In the Tool Positioning task, the task success is defined as whether the tool has been lifted and held in hand in a ready-to-use pose (e.g., hammer head facing down).

**Task objects.** In the Building Block task, we use the mesh model of Mega Blocks as our task objects. It is a range of large, stackable construction blocks designed specifically for the small hands of children. We take eight different types of blocks (denoted as Block 1, Block 2, . . . , Block 8). These blocks are illustrated in Appendix Figure. 6. In our experiment, Block 1-5 are used as the training objects and Block 6-8 are unseen ones for testing policy generalization. In the Tool Positioning task, the tools we consider consists of hammer, scoop and spatula, which have different thickness over the handle and variation of the center of mass. The hammer is used as a training object and the other two are unseen ones for testing policy generalization.

**Collision meshes.** We visualize the collision mesh used in the simulation in Figure. 7. We do observe the drop of simulation speed when loading 72 blocks, but it's still enough for training 1024 agents together at a speed of 5000 FPS with an NVIDIA RTX 3090 GPU. To optimize the speed, we reduce the resolution of the convex decomposition over blocks in Search, Orient and Grasp sub-tasks. The high-resolution blocks are only used for the training of the Insert sub-task.

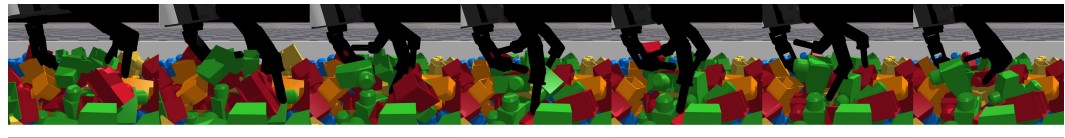

Figure 9: Snapshots of the searching task.

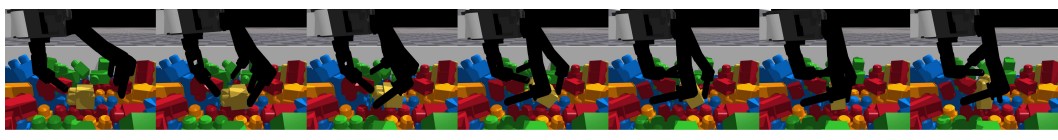

Figure 10: Snapshots of the orienting task.

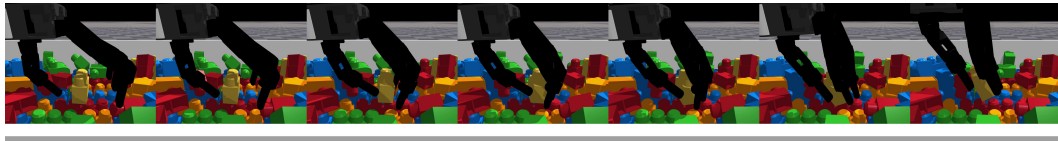

Figure 11: Snapshots of the grasping task.

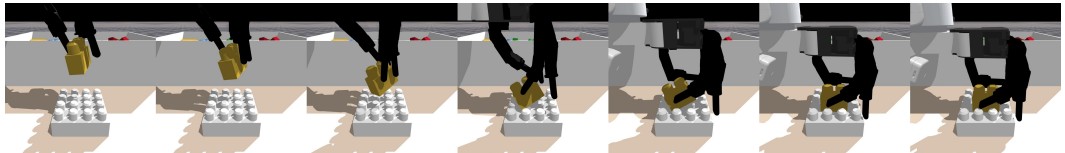

Figure 12: Snapshots of the inserting task.

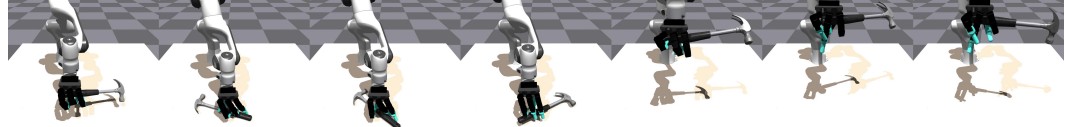

Figure 13: Snapshots of the hammer positioning.

### F.3   Tool positioning

For the tool positioning task, we have a total of three tools: hammer, spatula, and spoon. We use the hammer for training and test both in the hammer, spatula, and spoon. This long-horizon task involves grasp a tool and re-orient it onto a pose suitable for its use. Fig.8 shows what they look like and the initial and goal state of the each three tools.

### F.4   Typical frames of all sub-tasks

For the convenience of readers, we show some typical frames of all the sub-tasks in simulation.

### F.4.1   Building Blocks

We visualize the rollout of the Building Blocks task in Figure. 9, Figure. 10, Figure. 11, and Figure. 12.

### F.4.2   Tool Positioning

We visualize the rollout of the Tool Positioning task in Figure. 13, Figure. 14, and Figure. 15.

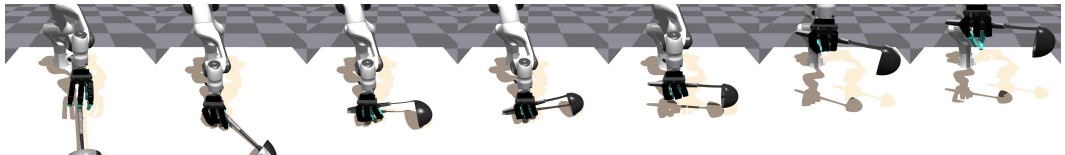

Figure 14: Snapshots of the spoon positioning.

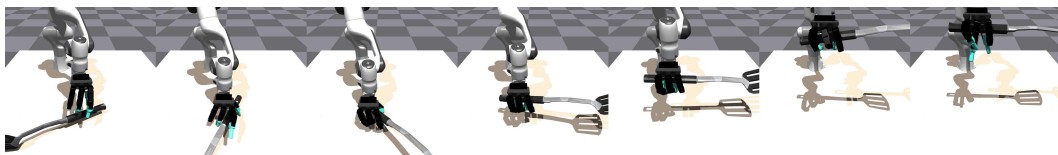

Figure 15: Snapshots of the spatula positioning.

|  | Trained | Unseen | All |
|---|---|---|---|
| Ours w/o belief state | $0.40_{\pm 0.08}$ | $0.16_{\pm 0.07}$ | $0.29_{\pm 0.06}$ |
| Ours w/o tactile | $\mathbf{0.43}_{\pm \mathbf{0.04}}$ | $0.33_{\pm 0.00}$ | $0.37_{\pm 0.02}$ |
| Ours w/o both | $0.26_{\pm 0.05}$ | $0.02_{\pm 0.01}$ | $0.14_{\pm 0.02}$ |
| Ours | $\mathbf{0.43}_{\pm \mathbf{0.04}}$ | $\mathbf{0.36}_{\pm \mathbf{0.04}}$ | $\mathbf{0.38}_{\pm \mathbf{0.04}}$ |

Table 8: Ablation study on the system choices in single-step **Orient** task

|  | Trained | | | | | Unseen | | | ALL |
|---|---|---|---|---|---|---|---|---|---|
|  | Block 1 | Block 2 | Block 3 | Block 4 | Block 5 | Block 6 | Block 7 | Block 8 | |
| Ours (0-step) | $0.47_{\pm 0.06}$ | $0.44_{\pm 0.07}$ | $0.43_{\pm 0.00}$ | $0.49_{\pm 0.04}$ | $0.40_{\pm 0.04}$ | $0.51_{\pm 0.04}$ | $0.18_{\pm 0.01}$ | $0.16_{\pm 0.03}$ | $0.38_{\pm 0.04}$ |
| Ours (5-step) | $0.52_{\pm 0.07}$ | $0.47_{\pm 0.02}$ | $0.46_{\pm 0.03}$ | $0.55_{\pm 0.03}$ | $0.44_{\pm 0.02}$ | $\mathbf{0.54}_{\pm \mathbf{0.01}}$ | $0.20_{\pm 0.03}$ | $\mathbf{0.17}_{\pm \mathbf{0.02}}$ | $0.42_{\pm 0.03}$ |
| Ours (10-step) | $\mathbf{0.61}_{\pm \mathbf{0.03}}$ | $\mathbf{0.55}_{\pm \mathbf{0.01}}$ | $0.52_{\pm 0.03}$ | $\mathbf{0.63}_{\pm \mathbf{0.03}}$ | $\mathbf{0.51}_{\pm \mathbf{0.06}}$ | $0.53_{\pm 0.06}$ | $\mathbf{0.22}_{\pm \mathbf{0.02}}$ | $0.16_{\pm 0.01}$ | $\mathbf{0.46}_{\pm \mathbf{0.03}}$ |
| Ours (15-step) | $0.55_{\pm 0.03}$ | $0.51_{\pm 0.04}$ | $\mathbf{0.53}_{\pm \mathbf{0.02}}$ | $0.59_{\pm 0.01}$ | $0.50_{\pm 0.07}$ | $0.50_{\pm 0.05}$ | $0.16_{\pm 0.04}$ | $0.14_{\pm 0.03}$ | $0.44_{\pm 0.04}$ |

Table 9: Ablation study in historical frame of the transition feasibility function

## G    Motor tactile and belief state.

We found that motor tactile and belief state are beneficial for dexterous in-hand manipulation. Tab. 8 is the ablation study of the design choices of our input state space. We modify the objective of the Orient sub-task in the Building Blocks task to a pre-defined goal orientation and train each ablation method only on this sub-policy. We find the belief state pose estimator has the highest improvement ($9\%$ in task success rate), which highlights its effects on in-hand manipulation.

## H    Ablation study in historical frame of the transition feasibility function

We add an ablation study by using 0-step, 5-step, 10-step and 15-step of history states as the inputs to the transition feasibility model, as shown in Table. 9. The task success rate gradually increases when more history steps are used and becomes stable after 10 steps. This result indicates that 10 to 15 history steps is ideal for the Building Block task.

## I    Environmental speed

Table. 10 shows the simulation FPS and wall-clock time cost of the training process for each sub-task. All of our experiments are run with Intel i7-9700K CPU and NVIDIA RTX 3090 GPU.

## J    Hyperparameters of the PPO

### J.1    Building Blocks

### J.2    Tool Positioning

| | Building Blocks | | | | Tool Positioning | |
|---|---|---|---|---|---|---|
| | Search | Orient | Grasp | Insert | Grasp | Reorient |
| Wall-clock time (s/10000 episode) | $31111_{\pm 3691}$ | $15458_{\pm 1381}$ | $16397_{\pm 1904}$ | $21851_{\pm 2791}$ | $17282_{\pm 2472}$ | $14500_{\pm 1831}$ |
| FPS (frame/s) | $1298_{\pm 154}$ | $5920_{\pm 529}$ | $5504_{\pm 639}$ | $14360_{\pm 1834}$ | $19920_{\pm 2849}$ | $20896_{\pm 2638}$ |

Table 10: Mean and standard deviation of FPS (frame per second) of the sub-tasks.

Table 11: Hyperparameters of PPO in Building Blocks.

| Hyperparameters | Searching | Orienting | Grasping & Inserting |
|---|---|---|---|
| Num mini-batches | 4 | 4 | 8 |
| Num opt-epochs | 5 | 10 | 2 |
| Num episode-length | 8 | 20 | 8 |
| Hidden size | [1024, 1024, 512] | [1024, 1024, 512] | [1024, 1024, 512] |
| Clip range | 0.2 | 0.2 | 0.2 |
| Max grad norm | 1 | 1 | 1 |
| Learning rate | 3.e-4 | 3.e-4 | 3.e-4 |
| Discount ($\gamma$) | 0.96 | 0.96 | 0.9 |
| GAE lambda ($\lambda$) | 0.95 | 0.95 | 0.95 |
| Init noise std | 0.8 | 0.8 | 0.8 |
| Desired kl | 0.016 | 0.016 | 0.016 |
| Ent-coef | 0 | 0 | 0 |

Table 12: Hyperparameters of PPO in Tool Positioning.

| Hyperparameters | Grasping | In-hand Orienting |
|---|---|---|
| Num mini-batches | 4 | 4 |
| Num opt-epochs | 5 | 10 |
| Num episode-length | 8 | 20 |
| Hidden size | [1024, 1024, 512] | [1024, 1024, 512] |
| Clip range | 0.2 | 0.2 |
| Max grad norm | 1 | 1 |
| Learning rate | 3.e-4 | 3.e-4 |
| Discount ($\gamma$) | 0.96 | 0.96 |
| GAE lambda ($\lambda$) | 0.95 | 0.95 |
| Init noise std | 0.8 | 0.8 |
| Desired kl | 0.016 | 0.016 |
| Ent-coef | 0 | 0 |

