# OpenReview forum: "Sequential Dexterity: Chaining Dexterous Policies for Long-Horizon Manipulation"
_robot-learning.org/CoRL/2023/Conference — CoRL 2023 Poster_

### Official Review · Reviewer_7DGx · 2023-07-12

**Confidence:** 5
**Originality:** Excellent
**Technical Quality:** Excellent
**Clarity Of Presentation:** Excellent
**Impact:** 4

**Recommendation:**

Strong Accept: I recommend accepting the paper and will argue for my recommendation even if other reviewers hold a different opinion.

**Review:**

This work proposes a novel and effective solution to long-horizon dexterous manipulation tasks while demonstrating impressive real world results via sim2real transfer. There are some minor weaknesses and points to be addressed, but overall this work clearly advances the state of the art for dexterous manipulation by showing providing a concrete method and result for extending dexterous manipulation to long-horizon settings.

Strengths:
* strong real world results involving long-horizon block assembly using dexterous hands
* thorough empirical evaluation in simulation against a variety of relevant baselines
* backward initialization is a neat technique for chaining sub-policies

Weaknesses:
* a crucial limitation is not mentioned: to use this method one has to define the sub-tasks, as well as their rewards, and a high-level plan (ordering of the subtasks)
* overall success rates are still fairly low (30%) for long-horizon tasks - room for improvement
* writing is missing some clarity (see questions)




**Quality Of The Limitations Section:**

Additional details required

**Questions For Rebuttal:**

Could the overall chained policy system be distilled into a single policy for simplified execution?

Why are the simulated results not near perfect? Given that these tasks can be trained on first billions of time steps - what is now the limiting factor - is it simply additional training time?

How is the target object's 6D pose estimated when it is in the middle of the bin - specifically for the search phase.

Missing details: can overall number of samples+wall-clock time (for specific machines used) be provided to get a sense of how long it takes to train the overall system

Clarity: Can it be made clear in the paper what the input / outputs to the system are? Having read through the paper + appendix, I believe the policies take in pose estimates and output joint position commands on the hand. In general it would be nice to have this information clearly spelled out (for both sim and real) in the main paper instead of having the reader dig through to find the details.

line 296 needs to be re-written (possible typo): "Second, our design choice of motor tactile does not yield a significant improvement in performance"

**Robotics Focus:**

Sufficient demonstration on hardware

**Summary Of Paper:**

The authors propose a method for performing long-horizon dexterous manipulation by sequentially training and finetuning sub-policies in simulation to be consistent across long-horizons. They then transfer these policies to the real robot to solve lego block assembly tasks. The paper proposes a new method for long-horizon chaining involving a finetuning process that enables the sub-policies to be effectively chained.

**Summary Of Recommendation:**

Overall, I strongly recommend acceptance. This work proposes a novel approach to long-horizon dexterous manipulation and demonstrates first of its kind results on block assembly with a multi-fingered hand. It would be great if the authors can address some of the minor points/clarifications that I have brought up.

---

> ### Author Response · Authors · 2023-08-15
> **Rebuttal Check-in**
>
> Thank you again for providing very constructive feedback. Since the end of rebuttal period is approaching (**11:59 PM PT on Aug 15**), we would like to kindly follow-up to check if the provided responses have sufficiently addressed your concerns. Please feel free to let us know if you have any other questions. Thanks again! :)

---

### Official Review · Reviewer_8hyr · 2023-07-16

**Confidence:** 5
**Originality:** Good
**Technical Quality:** Good
**Clarity Of Presentation:** Fair
**Impact:** 3

**Recommendation:**

Weak Reject: I recommend rejecting the paper, but will not argue for my recommendation if the majority of other reviewers have a different opinion.

**Review:**

**Strength**:
- Tackling a sequence of dexterous manipulation tasks is challenging yet important. It's good to see the authors' effort in this.
- The paper demonstrates some simulation-to-reality transfer, which is encouraging to see. Despite the preliminary real-world results, including physical experiments with a robot is a plus.

**Weaknesses**

- For equation (1), is the learning objective an MSE loss? Or is it just the difference, as shown in the paper?

- The reason the authors use 10 steps to predict the transition feasibility function is that the finger velocity cannot be captured in the state $s_t$. Why can't the state $s_t$ include the finger velocity? Even on a real robot, such information (joint velocity or fingertip velocity) is readily available.

- For the transition feasibility function, why not use just a binary indicator of whether the following subtask succeeds or fails? Correspondingly, in equation (2), one could simply give more positive rewards for subtask success and some negative penalty for subtask failure. This would be much simpler. If done this way, it would also make it easier to identify when to switch to the next primitive (Section 4.3), as it would just be checking a binary signal. With the current design in Section 4.3, one would have to manually tune the threshold for each task. Doesn't it seem that both things discussed in Section 4.3 (checking when to move to the next primitive or whether to skip it) could be achieved by just checking binary success/failure signals? I would love to see this simple binary success baseline included in the paper to demonstrate the benefit of using the return as the transition feasibility function.

- In Section 4.3: There are many prior works in the field of task and motion planning that consider when to switch from one primitive to another. The listed prior works in this section do not comprehensively cover the relevant literature. Identifying optimal times to switch primitives has been studied in prior works, such as in [3] and [4].

- Regarding the use of a sequence of historical states in learning the transition feasibility function, an ablation study is needed to evaluate the sensitivity of the final performance with respect to the number of historical steps. Specifically, the authors should include an analysis of the performance as the number of history steps is varied, to show the impact of this hyperparameter.

- The paper lacks details on the exact training and testing setup, making it difficult to judge the difficulty of the tasks. For example, the authors should provide specifics such as the success criteria, shapes/initial poses/goal poses of the blocks, and other implementation details.

- The authors should report the performance of each learned subpolicy on the corresponding subtask. Providing these results would allow readers to better evaluate the subpolicy training and performance for each modular component of the overall method.

- The paper only reports results using success rate as the sole metric, but further performance characterization is needed. First, the criteria for success are not clearly described. Second, the subpolicy behaviors in the videos appear rather unnatural and inefficient (seemingly bordering on random motion). For example, the grasping policies take a significant amount of time, (over 20 seconds in the first video ('Inserting a red block') on the project website). I would like to see plots showing the distribution of completion times for each subtask, both in simulation and real-world. In addition, the current videos suggest the policies for subtasks like reorientation are not well-optimized?

- In the tool positioning task (Table 2), it is unclear why the RL baselines exhibit very poor performance. From the videos, even the proposed method seems to try to orient the hammer to the right orientation during the initial grasping stage (not the reorientation stage), suggesting the reorientation policy is not contributing much. This raises questions about why the RL baselines cannot learn similar behaviors of properly orienting the tool while grasping. The authors should provide insight into why there is such a significant performance gap with the baselines on this specific task.

- The real-world object pose tracking in the videos appears quite noisy and unstable. It is unclear how the learned policies, especially for reorientation and grasping, work with such poor pose estimation. The paper states (Section B in appendix) that the insertion policy adjusts orientation based on estimated object pose. However, the videos show the object pose estimate fluctuating drastically while the fingers take no corrective action during insertion. This seeming contradiction between the described approach and observed behavior should be addressed. The authors need to discuss how the policies are robust to noisy tracking.

- Based on the real-world experiment videos, it seems that only the orientation and grasping behaviors use the learned policies, while the other stages are scripted or follow hand-coded logic. Could the authors clarify which parts of the full task demonstration utilize the learned policies versus pre-programmed actions?

- Missing details on the amount of training time for each subtask and forward-backward tuning with the transition feasibility function, how many forward-backward iterations are needed.

- The subtasks are not clearly defined in the main paper. For example, it was not until I read the whole appendix that I realized that the reorientation policy is simply attempting to reorient an object to a fixed goal orientation (should have been clarified in the reward function).


**Writing issues**

(1) The motivation behind using a dexterous hand, as explained in the first paragraph of the Introduction, is not well-grounded. The requirements for searching, reorienting, grasping, inserting, switching between different modes, and long-horizon planning do not necessarily justify the need for a dexterous hand. A parallel-jaw gripper, if implemented well, can potentially perform all of these tasks just as well (consider extrinsic dexterity as an example).

(2) Some of the words used to describe the results and contributions appear exaggerated or inaccurate. Some example are as follows:

The 'seamless zero-shot transfer' claim is questionable given the significant performance gap between simulation and the real world. The authors should tone down such claims.

Stating this is 'the first to explore long-horizon dexterous manipulation with policy chaining' overlooks relevant prior work like [1], which should be cited in Related Work as well.

Remove the words such as "optimal". There is no guarantee that the proposed method can find the "optimal" switch time. In fact, there might be no need to identify the "optimal" switch time.

Claiming that the proposed method can generalize to novel object shapes is an overstatement when the training and testing objects are limited to blocks.

(3) The claimed critical observation (page 5, Section 4.1) that "the successful end state of prior sub-task $\pi^{i-1}$ inherently provides plausible initial states for  $\pi^{i}$ to start with." was used by many prior works in long-horizon task planning. Even in dexterous manipulation, [2] used the end state distribution of a grasping policy as the initial state distribution of a in-air object reorientation policy. But such papers are not discussed/referenced in the section.

(4) The paragraph describing the real-world results needs to be revised. The current writing implies that no prior works are capable of solving this task and that only the proposed method can achieve a 33% success rate. However, it is not certain that other approaches cannot solve this problem with enough effort. There are multiple ways to tackle the issue. Therefore, it would be more accurate to rephrase the authors' results in a more cautious and precise manner. Additionally, having a 33% success rate is not sufficient to claim that the proposed method has "robustness".





[1]. Gupta, Abhishek, Justin Yu, Tony Z. Zhao, Vikash Kumar, Aaron Rovinsky, Kelvin Xu, Thomas Devlin, and Sergey Levine. "Reset-free reinforcement learning via multi-task learning: Learning dexterous manipulation behaviors without human intervention." In 2021 IEEE International Conference on Robotics and Automation (ICRA), pp. 6664-6671. IEEE, 2021.

[2]. Chen, Tao, Jie Xu, and Pulkit Agrawal. "A system for general in-hand object re-orientation." In Conference on Robot Learning, pp. 297-307. PMLR, 2022.

[3]. Su, Zhe, Oliver Kroemer, Gerald E. Loeb, Gaurav S. Sukhatme, and Stefan Schaal. "Learning to switch between sensorimotor primitives using multimodal haptic signals." In From Animals to Animats 14: 14th International Conference on Simulation of Adaptive Behavior, SAB 2016, Aberystwyth, UK, August 23-26, 2016, Proceedings 14, pp. 170-182. Springer International Publishing, 2016.

[4]. Kroemer, Oliver, Christian Daniel, Gerhard Neumann, Herke Van Hoof, and Jan Peters. "Towards learning hierarchical skills for multi-phase manipulation tasks." In 2015 IEEE international conference on robotics and automation (ICRA), pp. 1503-1510. IEEE, 2015.

**Quality Of The Limitations Section:**

Additional details required

**Questions For Rebuttal:**

- Regarding the state space, what exactly does "motor tactile" refer to? Additionally, in Section C, what is meant by the "number of pixels of the object exposed under the camera"?

- In Section D1, why is there a $\min$ operation in the reward function for searching? This would mean that there is no incentive for the fingers to get close to the object if they are already far away. Similarly, to maximize the reward, it would be best for the fingers to stay away from the object. The same question applies to the reward functions for grasping, reorientation, and insertion.

For the reward function of reorientation, what does the angle of rotation of the object mean? What is the goal of reorientation?

What is the motivation for using an exponential map in the grasping reward function, but not for the same term in the other two reward functions? How is the angular position difference between the object and the goal computed?

From the provided real-world videos, it does not seem that the insertion part is using any learned functions. Is it more likely using a scripted controller? So is the insertion policy actually being learned?

For the insertion reward, the coefficient for the position difference is $\alpha_0=1$, and the coefficient for the orientation difference is $\alpha_1=20$. Typically, the difference between orientations has a larger order of magnitude than the difference between positions. For example, a 0.4 rad difference in orientation is relatively small, but a 0.4 m difference in position is big. So why would the coefficient for the orientation difference be set to be even much bigger?

- How do the authors represent the state of other blocks (not the target block)?
- To learn the transition feasibility function, the authors regress it to the expected sum of rewards of the following subtasks. How is this done? Is it coming from Monte Carlo Sampling? How much variance does this have?
- Can you show pictures of the collision meshes used in the simulation? What's the fps of data generation in simulation? Since there are many lego blocks in the simulation, I assume the simulation will be slow?
- In the simulation videos, we can see a lot of high-frequency finger motion (jittering). However, in the real-world robot, I do not observe such jittering motion to the same extent. Can the authors elaborate on why this is the case?

- For insertion, how do the authors get the information of where to insert the block? Is it randomized during training and testing?

**Robotics Focus:**

Sufficient demonstration on hardware

**Summary Of Paper:**

The paper attempts to tackle the challenge of long-horizon dexterous manipulation. Specifically, the authors study the task of searching for a Lego block in a bin, reorienting it to the correct orientation, picking it up, and inserting it onto another Lego block. The sequence of subtasks is interesting. The main idea is to chain together dexterous skills for different subtasks by learning feasibility transition functions. These functions are then incorporated into the reward to maximize the chance of successfully completing the following subtask. While the paper provides a good starting point for this task formulation, substantial additional effort is needed to clearly scope the work and convey exactly what has been achieved.

**Summary Of Recommendation:**

Overall, the paper tackles an interesting problem and shows potential in how to solve it. However, there are many issues that need to be addressed, as detailed above. In addition, based on the initial submission, I am not confident that others would be able to reproduce the results due to questions surrounding the reward function, task setups, and other factors.

---

> ### Comment · Reviewer_8hyr · 2023-08-16
> **Post rebuttal**
>
> I appreciate the authors' response. A few more comments:
>
> - Towards the end of the introduction, the writing in the paper appears to suggest that both multi-stage policies were transferred to the real world. However, unless I missed it, the authors only transferred one of the tasks.
> - In terms of completion time, policies appear to perform better in the real world than in simulation. There are a few subtasks where completion time is shorter in the real world. However, it is difficult to be fully convinced of this due to the well-known sim-to-real gap.
> - In regards to the RL baseline for the tool positioning task, it is unclear which reward function (specifically which equation) the authors are referring to (where are alpha, beta, and gamma). And I am not convinced that the RL baselines require more reward tuning. The paper also utilizes RL with heavily engineered reward functions, which were likely manually tuned extensively in order to achieve the results. Therefore, it is not entirely convincing to claim that the baselines require more tuning.
> - In my opinion, claiming generalization to novel shapes is still an overstatement. At the very least, the authors should specify that it generalizes to novel shapes within the same object categories (LEGO blocks).
> - After comparing the two submissions, it seems that many of the answers in the rebuttals have not yet been incorporated into the paper. The authors should go through the answers to each question again and try to include them in the paper, as other readers may have similar questions.

---

> > ### Author Response · Authors · 2023-08-16
> > **Response to Comments**
> >
> > Thanks for the comments! Please find our response below:
> >
> > **1. “Towards the end of the introduction, the writing in the paper appears to suggest that both multi-stage policies were transferred to the real world. However, unless I missed it, the authors only transferred one of the tasks.”**
> >
> > Re: We would like to clarify that the last few sentences of the introduction draw two main conclusions: (1) Our approach outperforms baseline methods in both tasks. (2) Our method is able to sim-to-real transfer to a real-world dexterous robot system. In this work, we mainly showcase the sim-to-real transfer result in the more challenging Building Block task. Since in this response we are not allowed to attach a new revised PDF file, we will remove the word “two” in the last sentence in the next paper version.
> >
> > **2. “In terms of completion time, policies appear to perform better in the real world than in simulation. However, it is difficult to be fully convinced of this due to the well-known sim-to-real gap.”**
> >
> > Re: In this rebuttal experiment Q8, the completion time of the simulation result is the a combination of model inference and video rendering, which is heavily based on the rendering hardware (CPU and GPU of the workstation) and is not numerically comparable to the time cost in the real-world rollouts. One of the key aspects (in terms of time) that affects sim-to-real policy transfer is the control frequency (instead of time spent), which is the same between sim and real in our work as is introduced in Line 252-253.
> >
> > **3. “It is unclear which reward function (specifically which equation) the authors are referring to (where are alpha, beta, and gamma). And I am not convinced that the RL baselines require more reward tuning.”**
> >
> > Re: In this rebuttal experiment Q10, the alpha, beta and gamma are the weights between three reward terms introduced in Appendix Section D.2 for training the vanilla-RL, which individually encourages the policy to approach, lift and rotate the tool to the goal pose. Different from end-to-end vanilla RL, our method separates the training for each skill which alleviates the effort of tuning these weight terms. We agree reward tuning is an important step for all RL-based approaches, but as we discussed the method section 4.4, we empirically find that for learning long-horizon dexterous tasks, separate learning skills and use our bi-directional optimization to chain the policies cost less tuning effort and provide more stable results (across 5 random seeds as is shown in Table 1 and Table 2).
> >
> > **4. “In my opinion, claiming generalization to novel shapes is still an overstatement. At the very least, the authors should specify that it generalizes to novel shapes within the same object categories (LEGO blocks).”**
> >
> > Re: We agree that in the Building Blocks task, our generalization to novel shapes is within the category of LEGO blocks due to the design of the task. That's why we also investigate the Tool Positioning task and test the model learned with hammer-only on unseen tools such as scoop and spatula, which varies in thicknesses and center of mass. Generalization to new categories of objects is indeed an interesting future research direction we want to explore, but this topic is orthogonal to the main focus of this work (policy-chaining).
> >
> > **5. "After comparing the two submissions, it seems that many of the answers in the rebuttals have not yet been incorporated into the paper. The authors should go through the answers to each question again and try to include them in the paper, as other readers may have similar questions."**
> >
> > Re: Thanks for pointing out. We have carefully gone through the rebuttal again and find the response of Q3, Q7 ,Q8 and Q10 have not been included in the paper. Since we cannot attach new PDF to this response, we will include them in the next paper version. We appreciate your suggestions.
> >
> > We appreciate the thoroughness of the review and did our best to respond to all the concerns. We are happy to continue to work constructively and please feel free to let us know if you have any questions.

---

### Official Review · Reviewer_JYyi · 2023-07-21

**Confidence:** 4
**Originality:** Good
**Technical Quality:** Good
**Clarity Of Presentation:** Very Good
**Impact:** 3

**Recommendation:**

Weak Accept: I recommend accepting the paper, but will not argue for my recommendation if the majority of other reviewers have a different opinion.

**Review:**

The authors propose a reasonable approach to incentivizing succesful skill chaining and demonstrate its effectiveness convincingly.
They evaluate the approach with both simulation and real-world (via sim2real transfer) on a single-arm Franka Panda robot equipped with an allegro dexterous hand. Their proposed method dramatically outperforms end-to-end RL (both with and without a ciriculum) which they find to not work at all, and convincingly outperforms the tested baselines which include regularizing terminal policy states and attempting to add a chaining objective directly to skill reward functions.

Before moving on to discussion of the paper's limitations, I want to state clearly that this work is high-quality and clearly contributes to the field. There is dearth of work on long-horizon behavior with dexterous hands on real hardware and it's encouraging to see development of approaches that can produce reasonable results in that area. Furthermore, the authors chose an interesting experimental domain and reasonable baselines.

With that being said, there are still clearly limitations to this approach, the biggest one being a predetermined skill order. In more complex real-world tasks, execution order will be highly dependent on closed-loop interaction with the world. If something goes wrong, for instance the robot drops something, it will be necessary to sequence potentially unanticipated behaviors in order to recover. This is to say nothing of the multitask scenario where skills may need to be sequenced in may different ways depending on the high-level task being requested. As a result, I'm not convinced that a fixed forward-backward approach makes sense in the long term and this paper does not explicitly acknowledge that challenge.

I believe the current experiments could be further strengthened by an additional baseline as well: learn a fixed set of low-level policies independently of each other (with some given initial distribution for each) and then attempt the backward direction of optimization on those policies.

**Quality Of The Limitations Section:**

Additional details required

**Questions For Rebuttal:**

- What aspects of this approach would extend to a multitask setting with large numbers of skills where the ordering was not fixed or clear a priori?

- How much data was required during the backwards pass? Would it be feasible to incorporate real-world data into this part of the process?

**Robotics Focus:**

Sufficient demonstration on hardware

**Summary Of Paper:**

This paper proposes a factored approach to learning long-horizon dexterous behavior. They propose learning a set of short-horizon dexterous behaviors (via PPO in simulation) and explicitly incentivizing them to chain successfully via a two-step optimization. In the first step, they train skills sequentially based on the expected order of execution, treating the empirical end-state distribution of skill n-1 as the initial-condition distribution from which to sample when learning skill n. In the second backwards step, they then reverse this process, learning a chaining success function between each skill and finetuning skills to better chain with their successors.


**Summary Of Recommendation:**

The paper is technically well executed and provides a contribution to the field.

---

> ### Author Response · Authors · 2023-08-15
> **Rebuttal Check-in**
>
> Thank you again for providing the very constructive feedback. Since the end of rebuttal period is approaching (**11:59 PM PT on Aug 15**), we would like to kindly follow-up to check if the provided responses have sufficiently addressed your questions and concerns. If so, we kindly hope that you might be willing to raise the level of your recommendation. Thanks again! :)

---

### Official Review · Reviewer_NBQ2 · 2023-07-25

**Confidence:** 4
**Originality:** Very Good
**Technical Quality:** Excellent
**Clarity Of Presentation:** Excellent
**Impact:** 4

**Recommendation:**

Strong Accept: I recommend accepting the paper and will argue for my recommendation even if other reviewers hold a different opinion.

**Review:**

Strengths:
- To my knowledge, this paper presents a novel approach for long horizon dexterous manipulation
- I think the paper tackles a very difficult and important problem in an interesting and novel way
- The transition feasibility function is very useful for even more general purpose settings
- I also think the backward finetuning is important and clever way to approach the problem of compounding error when chaining skills
- The results are very impressive, especially the zero shot performance in the real world on the lego tasks. The videos look great as well
- The paper is very well written, clear and easy to follow


Weaknesses:
- It would be good to see more analysis on the different assumptions made that may not hold in the real world (like having access to 6D pose, etc).
- It would be good to see the tool task performed in the real world as well
- Some discussion on general applications to long horizon tasks (beyond dexterous manipulation ones)
- Some discussion needed on how well it can chain an unseen ordering of sub-policies, how to add new skills, etc

**Quality Of The Limitations Section:**

Limitations are addressed clearly

**Questions For Rebuttal:**

- How well does this approach work with different levels of noise in the 6D pose of different objects in the real world?
- What are the challenges of moving to visual inputs rather than object states?
- What are the challenges of trying this approach in the real world on tools?
- How can this technique be applied to more general setups? Maybe in settings where the skills are required to be combined in different ways?



**Robotics Focus:**

Sufficient demonstration on hardware

**Summary Of Paper:**

This paper presents a policy learning scheme for long horizon dexterous manipulation with RL. This is a challenging problem since naively combining sub-policies can lead to many compounding errors. This paper proposes to first learn sub-policies. For each task after the first, the end state distribution of the previous policy is used for initialization (i.e. the block will start in the hand), etc. To tackle the compounding errors, a transition feasibility function is learned to further finetune the policies. This function takes as input a set of end states from a previous policy, and is  supervised based on the reward obtained by the next policy. Once learned on policy i, it is used to finetune policy i-1, and so on, as an auxilliary reward. This also helps as a switching function between skills. The tasks involve looking for, orienting, grasping and inserting small lego blocks, as well as a tool grasping and reorientation strategy, showing good performance in sim and real world also.

**Summary Of Recommendation:**

I think the method is novel, it solves an important problem and the results (especially the lego task) are very good. In general this is a promising approach, and I would thus recommend acceptance of this paper.

---

> ### Author Response · Authors · 2023-08-15
> **Rebuttal Check-in**
>
> Thank you again for providing very constructive feedback. Since the end of rebuttal period is approaching (**11:59 PM PT on Aug 15**), we would like to kindly follow-up to check if the provided responses have sufficiently addressed your concerns. Please feel free to let us know if you have any other questions. Thanks again! :)

---

> > ### Comment · Reviewer_NBQ2 · 2023-08-16
> > **Response to Rebuttal**
> >
> > Dear Authors,
> >
> > Thank you for the thorough rebuttal - it has addressed my concerns. I will maintain my "strong accept" score, as I believe this paper should be presented at CoRL. Thanks!
> >
> > Best regards,
> >
> > Reviewer NBQ2

---

### Decision · Program_Chairs · 2023-08-30

**Decision:**

Accept (Poster)

**Comment:**

Authors present a method for sequencing skills for performing dexterous tasks. Reviewers unanimously agree that the problem is important and that the paper is a good contribution to the field. However, as 8hyr points out -- authors have not discussed relevant work in some parts of the writing and echoed concerns around reproducibility. Authors provided a response and I encourage the authors to release their code. Overall, I believe this paper makes a good contribution -- at the same time I encourage the authors to address remaining concerns of the reviewers -- especially from 8hyr to make their work even stronger.